# Estimating Above-Ground Biomass of Potato Using Random Forest and Optimized Hyperspectral Indices

**Haibo Yang** [1] , **Fei Li** [1,*], **Wei Wang** [2] **and Kang Yu** [3]

1   Inner Mongolia Key Laboratory of Soil Quality and Nutrient Resource, College of Grassland, Resources and Environment, Inner Mongolia Agricultural University, Hohhot, Inner Mongolia 010011, China; hbyang@emails.imau.edu.cn
2   ULanqab Institute of Agriculture and Forestry Sciences, ULanqab, Inner Mongolia 012000, China; wlcbww@163.com
3   Department Life Science Engineering, School of Life Sciences, Technical University of Munich, 85354 Freising, Germany; kang.yu@tum.de
*   Correspondence: lifei@imau.edu.cn

**Abstract:** Spectral indices rarely show consistency in estimating crop traits across growth stages; thus, it is critical to simultaneously evaluate a group of spectral variables and select the most informative spectral indices for retrieving crop traits. The objective of this study was to explore the optimal spectral predictors for above-ground biomass (AGB) by applying Random Forest (RF) on three types of spectral predictors: the full spectrum, published spectral indices (Pub-SIs), and optimized spectral indices (Opt-SIs). Canopy hyperspectral reflectance of potato plants, treated with seven nitrogen (N) rates, was obtained during the tuber formation and tuber bulking from 2015 to 2016. Twelve Pub-SIs were selected, and their spectral bands were optimized using band optimization algorithms. Results showed that the Opt-SIs were the best input variables of RF models. Compared to the best empirical model based on Opt-SIs, the Opt-SIs based RF model improved the prediction of AGB, with $R^2$ increased by 6%, 10%, and 16% at the tuber formation, tuber bulking, and for across the two growth stages, respectively. The Opt-SIs can significantly reduce the number of input variables. The optimized Blue nitrogen index (Opt-BNI) and Modified red-edge normalized difference vegetation index (Opt-mND705) combined with an RF model showed the best performance in estimating potato AGB at the tuber formation stage ($R^2 = 0.88$). In the tuber bulking stage, only using optimized Nitrogen planar domain index (Opt-NPDI) as the input variable of the RF model produced satisfactory accuracy in training and testing datasets, with the $R^2$, RMSE, and RE being 0.92, 208.6 kg/ha, and 10.3%, respectively. The Opt-BNI and Double-peak nitrogen index (Opt-NDDA) coupling with an RF model explained 86% of the variations in potato AGB, with the lowest RMSE (262.9 kg/ha) and RE (14.8%) across two growth stages. This study shows that combining the Opt-SIs and RF can greatly enhance the prediction accuracy for crop AGB while significantly reduces collinearity and redundancies of spectral data.

**Keywords:** potato crops; biomass estimation; machine learning; vegetation indices

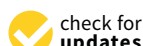



## 1. Introduction

Above-ground biomass (AGB) as an insightful indicator of crop production is essential to guide agricultural management practices [1–3]. Particularly, the AGB is the most important indicator for the calculation of the nitrogen nutrition index, which has been proposed to diagnose the plant nitrogen status for precise nitrogen fertilizer management in crops [4,5]. However, traditional manual measurements of the AGB are time-consuming and labor-intensive. These point-sampling-based methods are also infeasible in regional crop management [6–8]. Therefore, it is imperative to develop effective technologies to rapidly and accurately monitor crops AGB on a regional scale for precision crop nitrogen management [9,10].

Remote sensing is regarded as the most promising technology for timely and non-destructively acquiring crop canopy spatial variation [3,11]. The spectral parameters are widely used to estimate AGB at local, regional, and global scales [12–14]. For decades, spectral indices (SIs) are common parameters for the non-destructive estimation of AGB in crops [15,16]. Much research has reported that the SIs were used to estimate the AGB of maize [6,17,18], wheat [10,19,20], and rice [15,21]. However, the relationship between the SIs and AGB is inconsistent. For example, Venancio [17] found that the enhanced vegetation index (EVI), soil-adjusted vegetation index (SAVI), and optimized soil-adjusted vegetation index (OSAVI) were the best performing SIs for the estimation of maize AGB estimation, whereas Zhu [18] found that the normalized green red difference index (NGRDI) and simple ratio vegetation index (SRVI) had the highest correlation coefficient with AGB of maize. In contrast, Wang [6] found that the SIs showed poor relationships with maize AGB. The relationships between the selected SIs and crop properties are often inconsistent due to the influence of sites, years, and crop growth stages on the SIs algorithms and corresponding sensitive band combinations [22,23]. Therefore, the selection of suitable SIs algorithms and the sensitive band to develop the optimized SIs is necessary to guarantee the performance of SIs. Considering all possible band combinations according to established index formulations is a popular method to develop optimized SIs. The optimized SIs can improve the robustness and accurate of vegetation properties estimation by identifying optimal band combinations to some degree [24,25]. Many studies have demonstrated that it is an effective method for improving the estimation accuracy of AGB [21,26–28]. However, these SIs based methods keep on being restricted by using a few bands only. We could not ensure whether the spectral indices were the most effective indicators for estimating certain vegetation properties [29,30]. Although the band optimization can improve the robustness of SIs to some degree, the saturation effect is still a problem for quantifying moderate-high AGB in crops [31–34].

Recently, machine learning coupled with remotely sensed data have become a popular approach for vegetation parameters estimation [35,36]. Dayananda [37] used 121 hyperspectral bands from 450 to 998 nm coupling with a random forest algorithm to predict the biomass of maize with lower root mean square error. Similarly, the findings of Yu [38] confirmed that the partial least squares regression (PLSR) model could explain 91% variation of the rice AGB from the full spectrum analysis. However, the full spectrum has strong band continuity and higher dimensionality which reduce the computation efficiency and estimation accuracy of the model, especially for hyperspectral data [12,39,40]. Therefore, identifying the method to reduce the dimension of spectrum input variables is very important for machine learning algorithms. SIs are the mathematical transformation of several specific wavelengths. Much research has demonstrated that the SIs coupling with machine learning algorithms can achieve accurate estimation of crop biomass and other biophysical parameters [15,41]. For instance, Wang [41] predicted the wheat AGB using the support vector regression (SVR), the random forest (RF), and artificial neural network (ANN) based on 15 SIs, e.g., the normalized difference vegetation index (NDVI), optimized soil-adjusted vegetation index (OSAVI) and EVI. The SIs coupling with the SVR, RF, and ANN algorithms could explain 47–61%, 53–79%, and 30–49% of the variations in AGB at different growth stages, respectively. Niu [42] predicted the maize AGB using multivariable linear regression (MLR) based on the normalized green-red difference index (NGRDI), excess green minus excess red (ExGR) with an $R^2$ of 0.82. The partial least squares regression (PLSR) coupling with the two-band NDVI combinations estimated wheat AGB satisfactorily ($R^2 = 0.89$) [12]. In contrast, Wang [6] showed that the PLSR model coupling with SIs, e.g., NDVI, simple ratio vegetation index (SR), and modified soil-adjusted vegetation index (MSAVI), did not improve the biomass estimation accuracy in maize. Therefore, selecting suitable spectral variables as input parameters is critical for machine learning algorithms.

Machine learning offers a practical method for analyzing spectral information and accurately monitoring vegetation biophysical variables [29,30]. The previous studies have investigated the full spectrum and the SIs as input variables for machine learning to esti-

mate crop AGB. Spectral indices can effectively reduce the dimension and multicollinearity of machine learning input variables. However, to date, limited studies reported the Opt-SIs coupling with machine learning algorithms in the estimation of crop AGB. Among various machine learning methods, the RF algorithm has been regarded as one of the most commonly used prediction approaches for classification and regression due to the advantages of computation efficiency and insensitivity to over-fitting [41]. Therefore, the main objectives of the current research were (1) to evaluate the performances of published and Opt-SIs in estimating potato AGB, (2) to compare the performance of full-spectrum, Pub-SIs, and Opt-SIs coupled with RF models in predicting potato AGB.

## 2. Materials and Methods

### 2.1. Study Area and Experimental Design

Experiments were carried out in Wuchuan County, which is located in the middle of Inner Mongolia (extending from 110°31′E, 40°47′N to 111°53′E, 41°23′N), China, during the potato growing seasons of 2015 and 2016. As illustrated in Figure 1, the annual variations in temperature and precipitation are small. The area has a middle temperate arid and semi-arid continental monsoon, with cold winters and cool summers. The annual average precipitation is 398.3 mm and 90–95% of which occurs between April and October. The average temperature is 15.0 °C in the potato growing season. The main crops in this area are potato, sunflower, and oat.

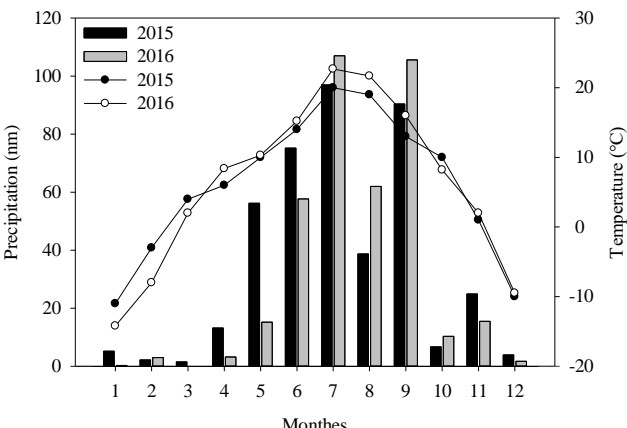

**Figure 1.** The variations of temperature and precipitation of Wuchuan County in 2015 and 2016.

Two field experiments with different N levels and one cultivar (Kexin 1) were conducted at two villages. Experiment 1was performed in Dong Liangshan with seven N rates (0, 83, 135, 165, 180, 250, and 424 kg N ha$^{-1}$) and four replications in 2015. The plot size was 9 m × 9 m. Nitrogen fertilizer applications with urea were split for before sowing and five growth stages, i.e., BBCH code 31, 40, 51, 60, and 70. At Dong Tucheng in 2016, Experiment 2 had seven N treatments, among the one control (0 kg N ha$^{-1}$), four optimum N treatments using urea, urea with urease inhibitor, urea ammonium nitrate solution, urea ammonium nitrate solution with a urease inhibitor, and two conventional N rates based on urea and urea ammonium nitrate solution. Each treatment had four replications and a plot of 8 m × 9 m. The N fertilizer was used as a fertigation split at five growth stages (BBCH 31, 40, 51, 60, and 70). The design of a completely random block was adopted in all experimental plots.

### 2.2. Data Collection

During the experimental periods, ground-based hyperspectral reflectance was measured at key growth stages of tuber formation and tuber bulking using a Field Spec Pro FR spectroradiometer (tec5, Oberursel, Germany). This instrument records reflectance between 300 and 1150 nm with a bandwidth resolution of 3.3 nm. As described by Li [22,27,32], the

measuring head of this device consists of two optics: the upper one is used to quantify the incoming light as a reference, and the lower one records the reflectance from the vegetation and ground with a 12° FOV. The bi-directional spectrometer was calibrated once with a Spectralon white reference panel. In the critical growing season of tuber formation and tuber bulking, which corresponded to the BBCH codes 40–49 and 60–69, potato canopy reflectance was measured by holding the sensor at nadir, from 0.5–0.8 m above the canopy, and walking a distance of 4–6 m with a constant speed along the potato ridges for each plot at a clear and windless day condition between 10:00 a.m. and 2:00 p.m. Subsequently, randomly selected two 1 m (1.8 m$^2$) consecutive rows of potato in the spectrometer-scanned area of each plot, all the above-ground plants were sampled and weighed the fresh weight. After chopping and mixing samples, 400–600 g sub-samples were oven-dried at 70 °C. to constant weight, after which the dry weight was determined. The AGB can be calculated as the following equation:

$$AGB \ (kg/ha) = \frac{FW}{1+M} / 1.8 \ m^2 \times 10000 \ m^2 \tag{1}$$

FW: fresh weight of all the above-ground plants.
M: moisture content of potato plants.

In the current studies, the sample of tuber formation and tuber bulking stages is 79 and 78, respectively. For each stage, the pooled data from 2015 and 2016 were randomly divided into a training dataset (70% of the pooled data) and an independent testing dataset (30% of the pooled data). For the training dataset, the number of samples was 55 at the tuber formation, 54 at the tuber bulking stage. For the test dataset, there were 24 samples for each growth stage. The pooled datasets (All) were derived from the combination of the data of the above two stages, including 109 samples in the training dataset and 48 samples in the testing dataset. The training dataset was used to establish models to investigate the performance of RF algorithm coupling with different spectral variables. The test dataset was used to test the accuracy and robustness of each prediction model.

### 2.3. Spectral Indices

Many SIs have been developed to estimating crop biophysical parameters. Among them, the two-band spectral indices simple ratio vegetation index [43], normalized difference vegetation index (NDVI) [44], and difference vegetation index [45] are the most classic spectral indices algorithms. Subsequently, the three-band spectral indices were proposed by adding new wavebands and changing the formula formats. For example, Sims and Gamon [46] modified the normalized difference (ND) formula formats by using blue reflectance ($R_{455}$) as a measure and proposed the new three-band spectral indices ND705. In the current study, we chose 12 Pub-SIs (Table 1) and used the "lambda-by-lambda" band-optimization algorithm which is widely used for the optimization of spectral indices [47–49] to determine the best band combination for different spectral indices formula formats. The Opt-SIs were used as predictors in the RF models for potato AGB estimation. More details regarding the process of selecting spectral index sensitive bands are described in our previous work [27,50,51].

### 2.4. Random Forest Regression

The RF regression model is an ensemble learning algorithm that combines a large number of decision trees (*ntree*). When each tree was built, two-third of the training samples were used to training the model, and one-third of training samples, called out-of-bag (OOB) samples were left out [3,52]. The prediction result was determined by averaging over all the trees [3,41,53].

RF algorithm has been regarded as one of the most accurate prediction approaches for classification and regression [41]. The framework parameters (e.g., *n_estimators*, *oob_score* and *criterion*) and decision tree parameters (e.g., *max_depth*, *max_features*, and *max_leaf_nodes*) are an important part of the RF algorithm. Among them, the number of decision trees

(*ntree*) and input variables per node (*mtry*) are the most frequently concerned parameters to improve operating efficiency and estimation accuracy. However, most of the studies have by far focused on the *ntree* and *mtry*, and have not recognized the importance of input variables to achieve their best performance. Therefore, three types of spectral input variables: full-spectrum, Pub-SIs, and Opt-SIs were applied in the current study to investigate the influence of spectral input variables on the estimation ability of random forest algorithm.

To evaluate the influences of different input variables coupling with the RF model to predict potato AGB, we set three decision tree levels: high (ntree = 5000), medium (ntree = 2000), and low (ntree = 500), and other parameters were set to default values according to the *scikit-learn* software package (https://scikit-learn.org/stable/modules/generated/sklearn.ensemble.RandomForestClassifier.html). Subsequently, the number of the mtry and ntree were further optimized according to the 'importance scores' and root square error (RMSE) for simplifying the prediction model and enhancing the operational efficiency and accuracy.

### 2.5. Variable Importance Score

The RF model can assess the variable importance score according to the out-of-bag (OOB) error estimates in the model [21]. Importance score as an evaluation indicator was widely used to select the input variables to simplify the RF model [52,54]. The importance score can be calculated as follow:

$$\text{Importance score} (X) = \sum_{i=1}^{n} \frac{\text{errOOB2} - \text{errOOB1}}{n} \tag{2}$$

where errOOB1 represents the error of out-of-bag for variable X with one decision tree, errOOB2 represents the error of adding noise to variable X with one decision tree, and n represents the number of decision trees.

### 2.6. Model Accuracy

The performances of the different models of RF and spectral indices were evaluated by comparing the coefficients of determination ($R^2$) [55] relative error (RE, %) [56] and root square error (RMSE) [57] in predictions. The higher the $R^2$ and the lower the RMSE and RE, the better the precision and accuracy of the models. The $R^2$, RMSE, and RE were calculated as following equations:

$$R^2 = \sum (y_i - \overline{y})^2 / \sum (y_i - \hat{y}_i)^2 \tag{3}$$

$$RMSE = \sqrt{\frac{1}{n} \sum_{i=1}^{n} (y_i - \hat{y}_i)^2} \tag{4}$$

$$RE(\%) = \frac{RMSE}{\overline{y}} * 100 \tag{5}$$

where $\hat{y}_i$, $y_i$, and $\overline{y}$ are the measured, predicted, and mean values of AGB, respectively, and n is the number of samples.

**Table 1.** Spectral indices used in this study.

| Types | Spectral Indices | Abbreviations | Formulas | Algorithms | References |
|---|---|---|---|---|---|
| Two-band spectral indices | Ratio vegetation index | RVI | $R_{800}/R_{670}$ | $R_{\lambda 1}/R_{\lambda 2}$ | [43] |
| | Normalized difference vegetation index | NDVI | $(R_{800} - R_{680})/(R_{800} + R_{680})$ | $(R_{\lambda 1} - R_{\lambda 2})/(R_{\lambda 1} + R_{\lambda 2})$ | [44] |
| | Different vegetation index | DVI | $R_{800} - R_{680}$ | $R_{\lambda 1} - R_{\lambda 2}$ | [45] |
| | Modified soil-adjusted vegetation index | MSAVI | $0.5 \times (2 \times R_{810} + 1 - ((2 \times R_{810} + 1) \times 2 - 8 \times (R_{810} - R_{670})) \times 0.5)$ | $0.5 \times (2 \times R_{\lambda 1} + 1 - ((2 \times R_{\lambda 1} + 1) \times 2 - 8 \times (R_{\lambda 1} - R_{\lambda 2})) \times 0.5$ | [58] |
| | The renormalized difference vegetation index | RDVI | $(R_{800} - R_{670})/\mathrm{sqrt}(R_{800} + R_{670})$ | $(R_{\lambda 1} - R_{\lambda 2})/\mathrm{sqrt}\,(R_{\lambda 1} + R_{\lambda 2})$ | [59] |
| | Optimal vegetation index | VIopt | $(1 + 0.45) \times ((R_{800}) \times 2 + 1)/(R_{670} + 0.45)$ | $(1 + 0.45) \times ((R_{\lambda 2}) \times 2 + 1)/(R_{\lambda 1} + 0.45)$ | [60] |
| | Canopy chlorophyll content index | CCCI | $(NDRE - NDRE_{MIN})/(NDRE_{MAX} - NDRE_{MIN})$ | $(NDRE - NDRE_{MIN})/(NDRE_{MAX} - NDRE_{MIN})$ | [61] |
| Three-band spectral indices | Modified red-edge normalized difference vegetation index | mND705 | $(R_{750} - R_{705})/(R_{750} + R_{705} - 2 \times R_{445})$ | $(R_{\lambda 1} - R_{\lambda 2})/(R_{\lambda 1} + R_{\lambda 2} - 2 \times R_{\lambda 3})$ | [46] |
| | Blue nitrogen index | BNI | $R_{434}/(R_{496} + R_{401})$ | $R_{\lambda 1}/(R_{\lambda 2} + R_{\lambda 3})$ | [62] |
| | Nitrogen planar domain index | NPDI | $(CI_{\text{green edge}} - CI_{\text{green edge MIN}})/(CI_{\text{green edge MAX}} - CI_{\text{green edge MIN}})$ | $(CI_{\text{green edge}} - CI_{\text{green edge MIN}})/(CI_{\text{green edge MAX}} - CI_{\text{green edge MIN}})$ | [51] |
| | Double-peak nitrogen index | NDDA | $(R_{755} + R_{680} - 2 \times R_{705})/(R_{755} - R_{680})$ | $(R_{\lambda 1} + R_{\lambda 2} - 2 \times R_{\lambda 3})/(R_{\lambda 1} - R_{\lambda 2})$ | [63] |
| | Modified red-edge ratio | mRER | $(R_{759} - 1.8 \times R_{419})/(R_{742} - 1.8 \times R_{419})$ | $(R_{\lambda 1} - 1.8 \times R_{\lambda 2})/(R_{\lambda 3} - 1.8 \times R_{\lambda 2})$ | [64] |

R: the abbreviation of reflectance. λ: the wavebands of spectral indices.

## 3. Results

### 3.1. Variations in Potato AGB

As illustrated in Figure 2, the potato AGB increased from tuber formation to tuber bulking stage. The AGB in the training dataset ranged from 492.8 kg/ha to 3881.0 kg/ha with a CV value of 48.1%, while it varied from 497.2 kg/ha to 3353.5 kg/ha with a CV value of 44.9% for the testing dataset during the tuber formation and tuber bulking stage. The training and testing datasets exhibited a similar statistical distribution of AGB, avoiding potentially biased estimations in model calibration and validation.

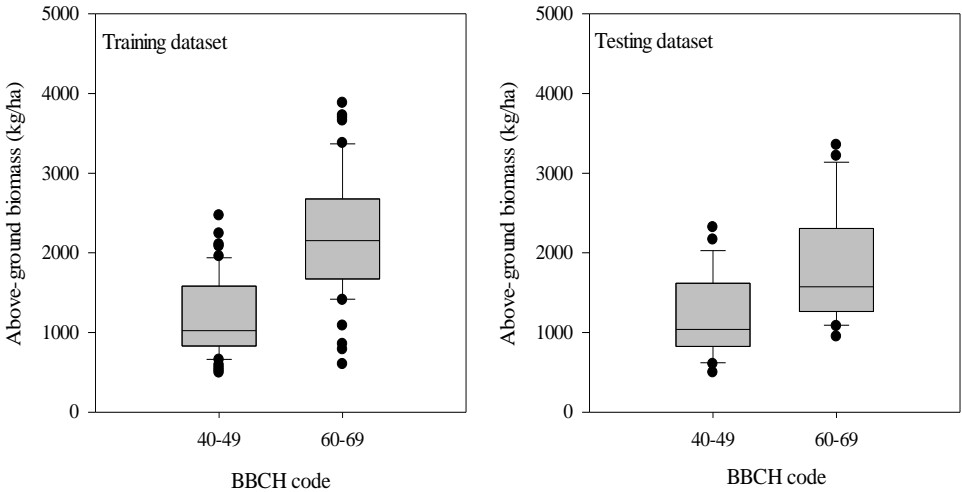

**Figure 2.** Variation of the above-ground biomass of potato at tuber formation and tuber bulking stages. The distribution is characterized by box-and-whisker plots, where the boxes show the 25th and 75th percentiles and the whiskers the 10th and the 90th percentiles. The median is represented by the line in the box and is provided as a number above the box plot.

### 3.2. Relationships between Spectral Indices and Potato AGB

Table 2 shows the relationships between Pub-SIs and potato AGB based on the training dataset. The results indicated that the Pub-SIs could explain 6–62% of the variations in potato AGB for different growth stages. The growth stages significantly affect the performances of Pub-SIs. The best performing Pub-SIs were mND705, CCCI, and RVI for the tuber formation stage, tuber bulking stage, and the combination of two stages, respectively.

**Table 2.** The band (nm) combination of published spectral indices (Pub-SIs) and corresponding relationships between Pub-SIs and potato AGB at different growth stages.

| Pub-SIs | Band Combinations | | | $R^2$ | | |
|---|---|---|---|---|---|---|
| | $R_{\lambda 1}$ | $R_{\lambda 2}$ | $R_{\lambda 3}$ | Tuber Formation | Tuber Bulking | All |
| RVI | 800 | 670 | | 0.36 | 0.43 | 0.62 |
| NDVI | 800 | 680 | | 0.28 | 0.34 | 0.52 |
| DVI | 800 | 680 | | 0.06 | 0.48 | 0.26 |
| MSAVI | 810 | 670 | | 0.27 | 0.32 | 0.50 |
| RDVI | 800 | 670 | | 0.13 | 0.48 | 0.39 |
| VLopt | 800 | 670 | | 0.28 | 0.49 | 0.61 |
| CCCI | 800 | 720 | 670 | 0.28 | 0.58 | 0.17 |
| mND705 | 750 | 705 | 445 | 0.52 | 0.53 | 0.47 |
| BNI | 434 | 496 | 401 | 0.31 | 0.53 | 0.31 |
| NPDI | 806 | 738 | 560 | 0.15 | 0.25 | 0.23 |
| NDDA | 755 | 680 | 705 | 0.50 | 0.52 | 0.22 |
| mRER | 759 | 419 | 742 | 0.46 | 0.56 | 0.44 |

Pub: the abbreviations of published. R: the abbreviations of reflectance.

The Opt-SIs significantly improved the performances compared to the Pub-SIs. The Opt-SIs explained 42–80% of the variations in potato AGB (Table 3). The predicting ability of three-band Opt-SIs outperformed the two-band Opt-SIs. The optimized NPDI, BNI, and mRER were the best performing Opt-SIs for the tuber formation stage, tuber bulking stage, and the combination of two stages, respectively. Compared with the Pub-SIs, the Opt-SIs had a great variation in the sensitive band combinations due to the influence of the growth stages (Tables 2 and 3). The sensitive bands were mainly located at ultraviolet (350–450 nm), blue (450–500 nm), and near-infrared (NIR, 800–1100 nm) at all three datasets.

**Table 3.** The band (nm) combination of Opt-SIs and corresponding relationships between Opt-SIs and potato AGB at different growth stages.

| Opt-SIs | Tuber Formation | | | | Tuber Bulking | | | | All | | | |
|---|---|---|---|---|---|---|---|---|---|---|---|---|
| | $R_{\lambda 1}$ | $R_{\lambda 2}$ | $R_{\lambda 3}$ | $R^2$ | $R_{\lambda 1}$ | $R_{\lambda 2}$ | $R_{\lambda 3}$ | $R^2$ | $R_{\lambda 1}$ | $R_{\lambda 2}$ | $R_{\lambda 3}$ | $R^2$ |
| RVI | 608 | 472 | | 0.59 | 1096 | 1094 | | 0.74 | 820 | 600 | | 0.66 |
| NDVI | 608 | 472 | | 0.59 | 1096 | 1094 | | 0.75 | 1020 | 936 | | 0.71 |
| DVI | 980 | 946 | | 0.62 | 1096 | 1094 | | 0.75 | 1014 | 1008 | | 0.67 |
| MSAVI | 944 | 728 | | 0.63 | 1096 | 1094 | | 0.75 | 1008 | 936 | | 0.72 |
| RDVI | 974 | 936 | | 0.68 | 1096 | 1094 | | 0.74 | 998 | 934 | | 0.73 |
| VLopt | 562 | 324 | | 0.42 | 772 | 304 | | 0.56 | 770 | 658 | | 0.61 |
| CCCI | 402 | 404 | 486 | 0.72 | 1096 | 1094 | 308 | 0.74 | 822 | 986 | 812 | 0.71 |
| mND705 | 492 | 386 | 412 | 0.72 | 1094 | 308 | 1096 | 0.76 | 998 | 934 | 1148 | 0.74 |
| BNI | 412 | 404 | 418 | 0.74 | 1094 | 416 | 1096 | 0.78 | 1020 | 908 | 1034 | 0.75 |
| NPDI | 362 | 444 | 458 | 0.76 | 1084 | 1096 | 1094 | 0.75 | 946 | 558 | 1020 | 0.72 |
| NDDA | 492 | 406 | 394 | 0.73 | 1096 | 308 | 1094 | 0.75 | 812 | 822 | 986 | 0.71 |
| mRER | 730 | 760 | 1140 | 0.73 | 498 | 1096 | 1094 | 0.80 | 932 | 1146 | 986 | 0.73 |

All: the combination of two growth stages.

Figure 3 shows the accuracy and precision of the AGB estimation based on the best performing Opt-SIs. The results showed that the Opt-SIs had the higher $R^2$ (0.39–0.81), lower RMSE (231.8–386.1 kg/ha), and RE% (14.1–25.4%) in different testing datasets compared to Pub-SIs.

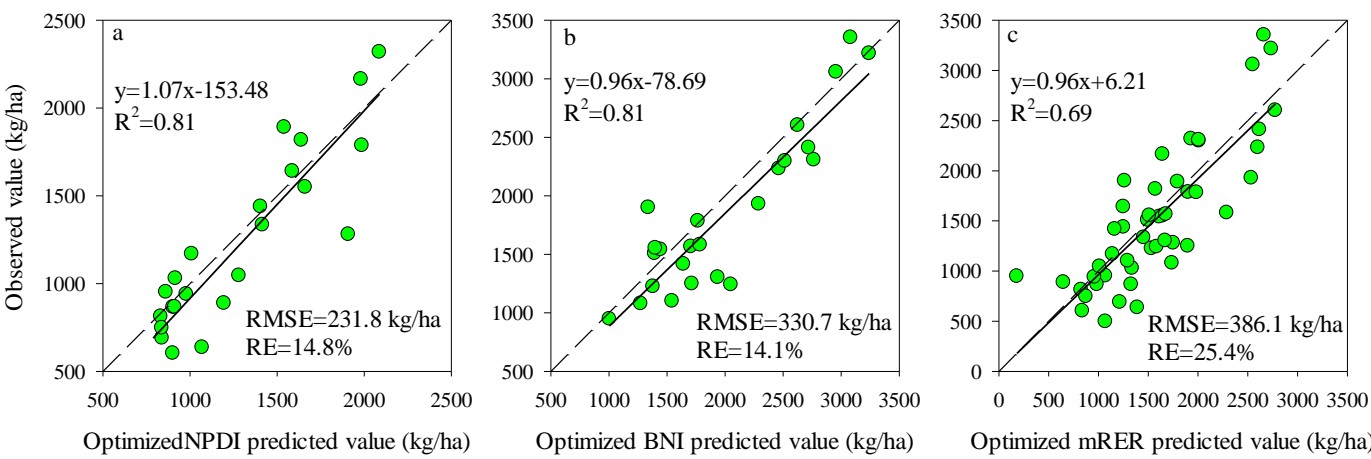

**Figure 3.** The relationships between the observed value and predicted value for the testing datasets using the best Opt-SIs at (**a**) tuber formation stage, (**b**) tuber bulking stage, and (**c**) the combination of growth stages.

### 3.3. Estimation of Potato AGB Using RF Model

Comparison analysis in the performance of RF model-based AGB estimation was conducted for the training dataset using different spectrum input variables. The relationships between observed and predicted AGB at different growth stages are presented in Figure 4. The input variables had a significant influence on the predicting ability of RF

models. Compared with the published and Opt-SIs, the full spectrum coupling with the RF model had the highest RMSE and RE% in the estimation of AGB. RF model coupled with the Opt-SIs had the best performances in different stages and decision trees. Using the Opt-SIs as the input variables of the RF model significantly improved the prediction of potato AGB in different training datasets. The number of regression trees, i.e., 500, 2000, and 5000, did not affect the performance of the models, indicating that the initial number of decision trees is adequate to explain the variation of AGB in the training dataset.

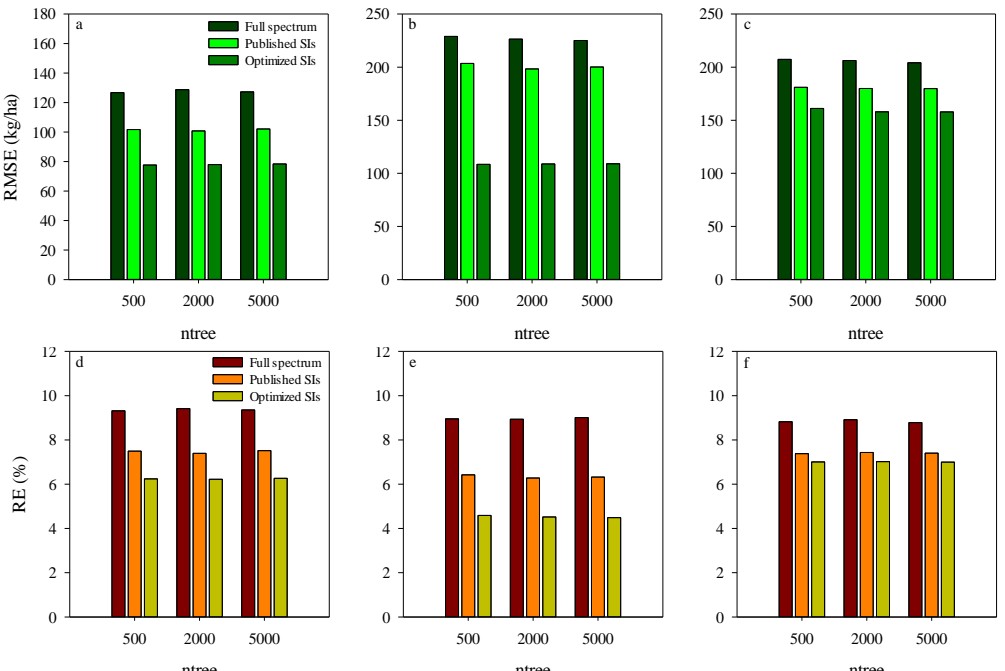

**Figure 4.** The RMSE and RE of training dataset using RF model based on different input variations at tuber formation stage (**a**,**d**), tuber bulking stage (**b**,**e**), and the combined stage of tuber formation and tuber bulking (**c**,**f**).

To further evaluate the performance of RF model coupling with different input variables, the estimation accuracy of RF models was validated with the testing datasets at each stage. The results showed that the RF models coupled with Opt-SIs had the best predicting performance in the estimation of potato AGB at all growth stages, with the highest $R^2$ (0.85–0.91), the lowest RMSE (192.2–273.4 kg/ha), and RE% (11.7–13.4%) (Figure 5a–c). Compared to the Opt-SIs, the full spectrum and Pub-SIs coupling with RF models showed the least performance at different growth stages, especially the Pub-SIs due to the influence of growth stages.

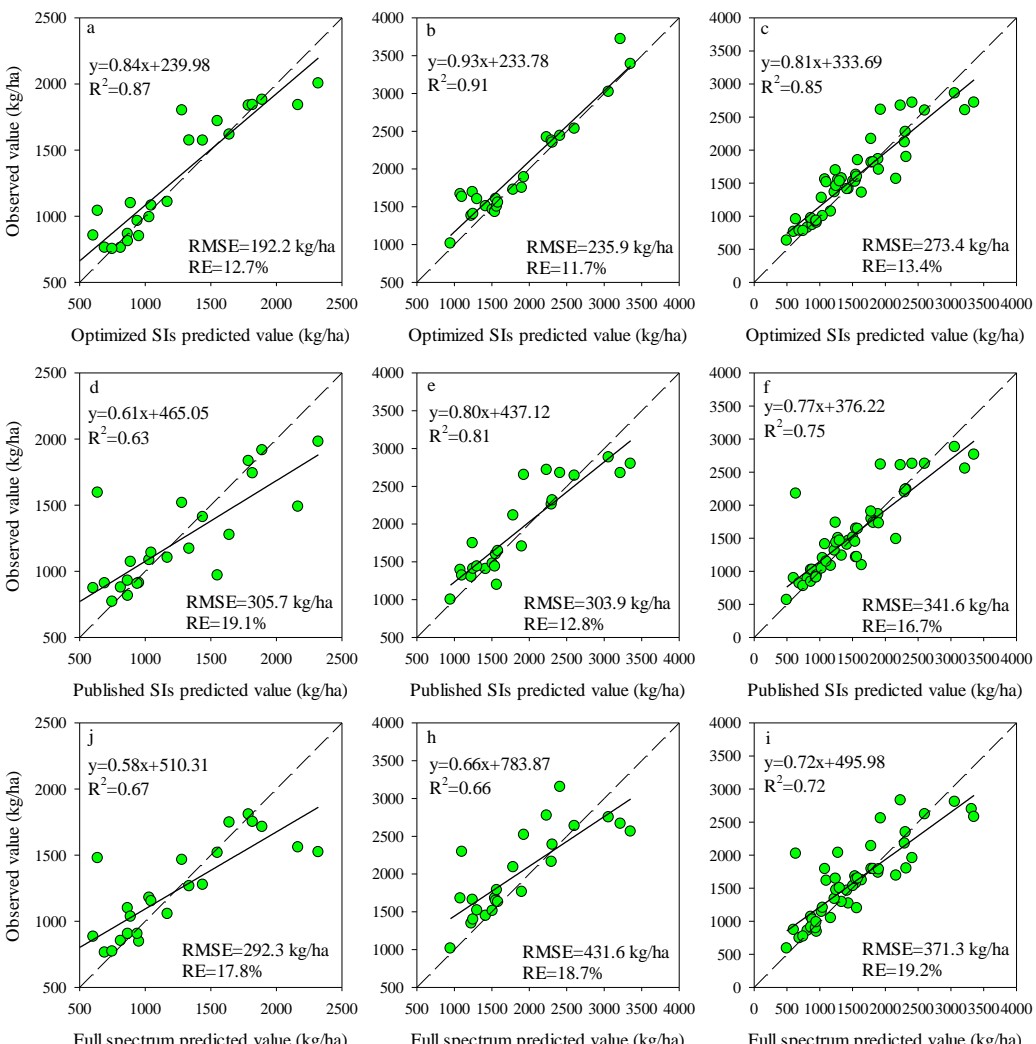

**Figure 5.** The relationships between the predicted value and observed value for different spectral variables of RF model at tuber formation (**a**,**d**,**j**), tuber bulking (**b**,**e**,**h**), and combined stage (**c**,**f**,**i**).

### 3.4. The Optimization of RF Model

To obtain the most efficient model for estimating potato AGB, the *ntree* and *mtry* were further optimized using the training dataset for the models of Opt-SIs and full-spectrum coupling with the RF algorithm. For each stage, the *mtry* values from 1 to 12 with an interval of length 1 were tested with the *ntree* values of 150, 300, and 500. The *ntree* and *mtry* values with the lowest RMSE were regarded as the best selection. According to Figure 6, the values of *ntree* and *mtry* were 350 and 5 at the tuber formation (RMSE = 74.9 kg/ha), 150 and 8 at tuber bulking (RMSE = 103.2 kg/ha), and 150 and 3 at the combination of two growth stages (RMSE = 150.8 kg/ha) were the best performing parameters for the models of Opt-SIs coupling with RF algorithm. Compared with the Opt-SIs, the training results of the RF model were poor, with the higher RMSE of 176.4–188.6 kg/ha, 211.0–230.1 kg/ha, and 122.0–134.0 kg/ha at tuber formation, tuber bulking and the combination of growth stage, respectively. The best optimized RF models were tested using the testing datasets and explained 85–91% variation of potato AGB with lower RMSE (185.4–273.5 kg/ha) and RE% (11.1–13.7%) (Figure 7).

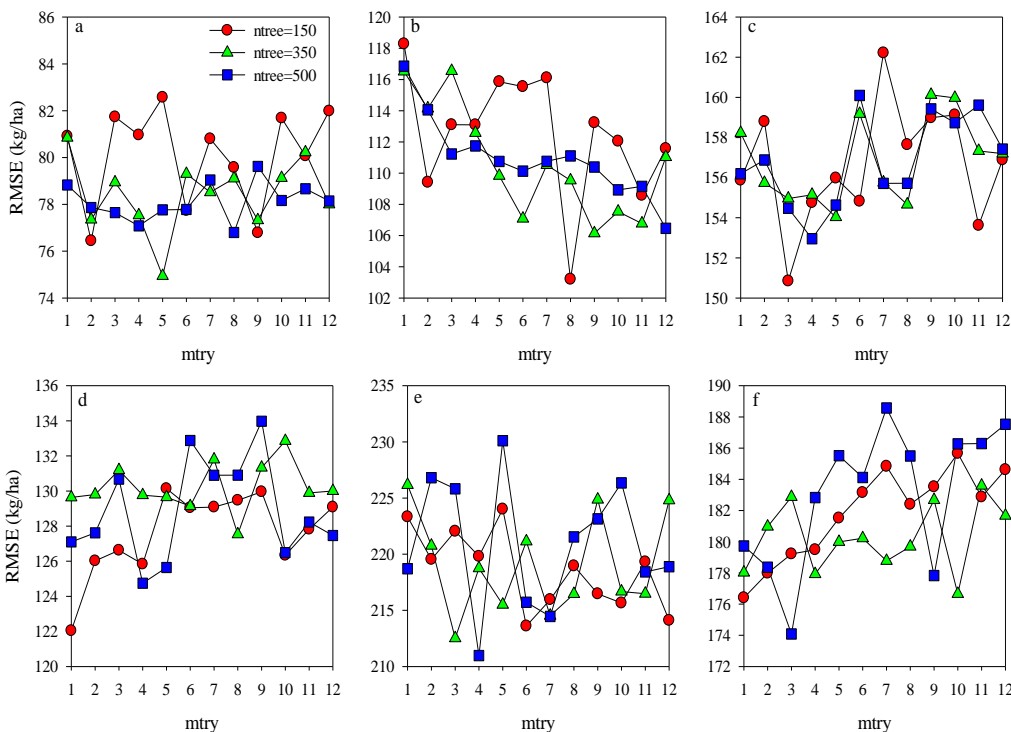

**Figure 6.** Optimization of random forest parameters (ntree and mtry) using RMSE at different growth stages for training dataset (**a**–**c** represent the Opt-SIs coupling with RF algorithm at (**a**) tuber formation, (**b**) tuber bulking, and (**c**) the combined stage; (**d**–**f**) represent the full spectrum coupling with RF algorithm at (**d**) tuber formation, (**e**) tuber bulking, and (**f**) the combined stage.).

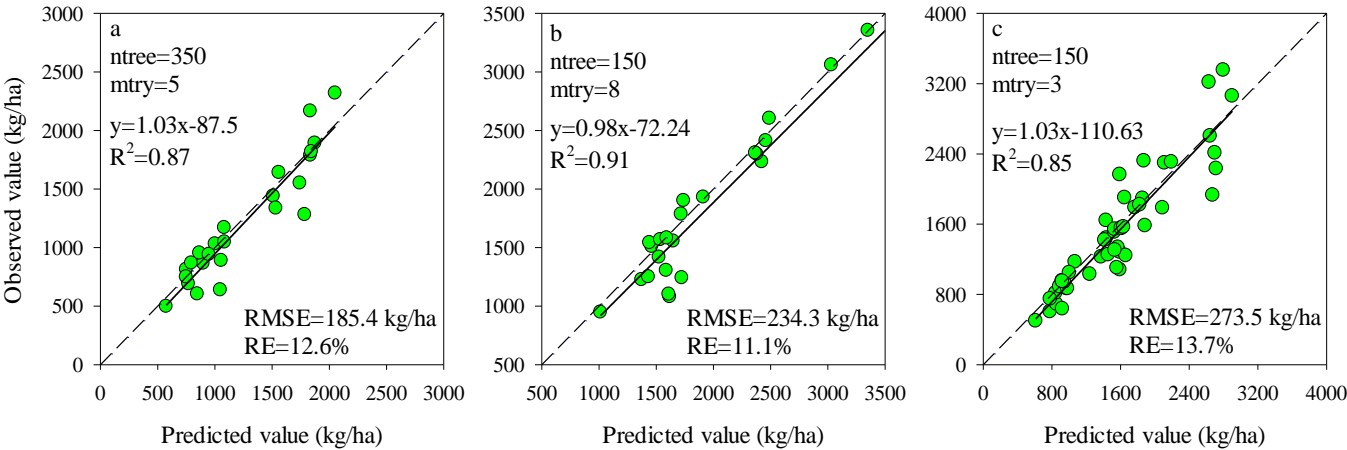

**Figure 7.** Predictive performance of RF models based on the best modeling in Figure 4 at (**a**) tuber formation, (**b**) tuber bulking, and (**c**) the combined growth stage using testing dataset.

The importance scores of predictors for RF models can be used to screen the input variables of prediction models. The importance scores of Opt-SIs in the best performing RF model were further summarized and analyzed and found that the importance of the Opt-SIs differed among models (Figure 8). The three-band-based Opt-SIs had relatively higher importance scores compared to two-band-based Opt-SIs. The number of model input parameters was optimized according to the importance scores. The results showed that the optimized BNI (412, 404, and 418 nm) and mND705 (492, 386, and 412 nm) coupling with the RF model achieved good prediction results in the training dataset and the testing dataset at the tuber formation stage (Figure 9). For the tuber bulking stage, the best prediction accuracy for potato AGB was achieved using only one predictor: optimized

NPDI, with sensitive bands located at 1084 nm, 1096 nm, and 1094 nm. The RF model based on optimized NPDI can explain 92% variation of potato AGB when the number of trees was 150, the RMSE and RE% were 208.6 kg/ha and 10.3%, respectively (Figure 10). Compared with a single growth stage, the optimized BNI (1020, 908, and 1034 nm) coupling with RF had better performance in the training dataset, while the RMSE, RE%, and $R^2$ were relatively high in the testing dataset at the combination of two growth stages (Figure 11a,d). However, using two or three Opt-SIs, e.g., optimized BNI (1020, 908 and 1034 nm), NDDA (812, 822 and 986 nm), and mND705 (998, 934, and 1148 nm) enables efficient and accurate prediction of potato AGB in both of the training and testing datasets (Figure 11b,c,e,f). Across the two growth stages, the optimized RF model could explain 86–87% variation of potato AGB.

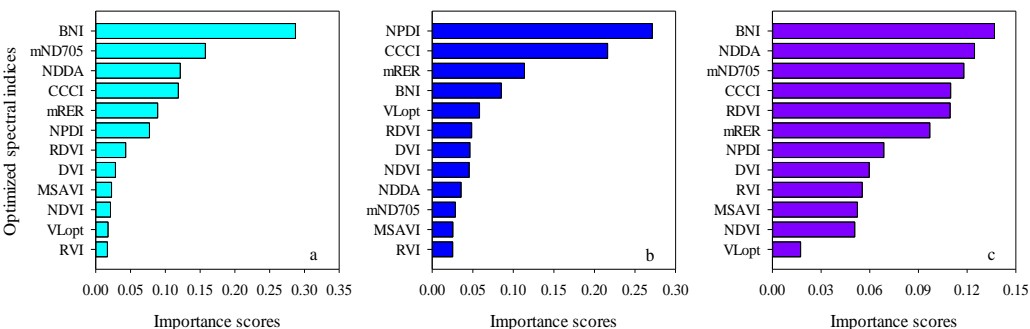

**Figure 8.** Importance scores both Opt-SIs in the best RF modeling for predicting potato AGB at (**a**) tuber formation, (**b**) tuber bulking, and (**c**) the combined growth stage.

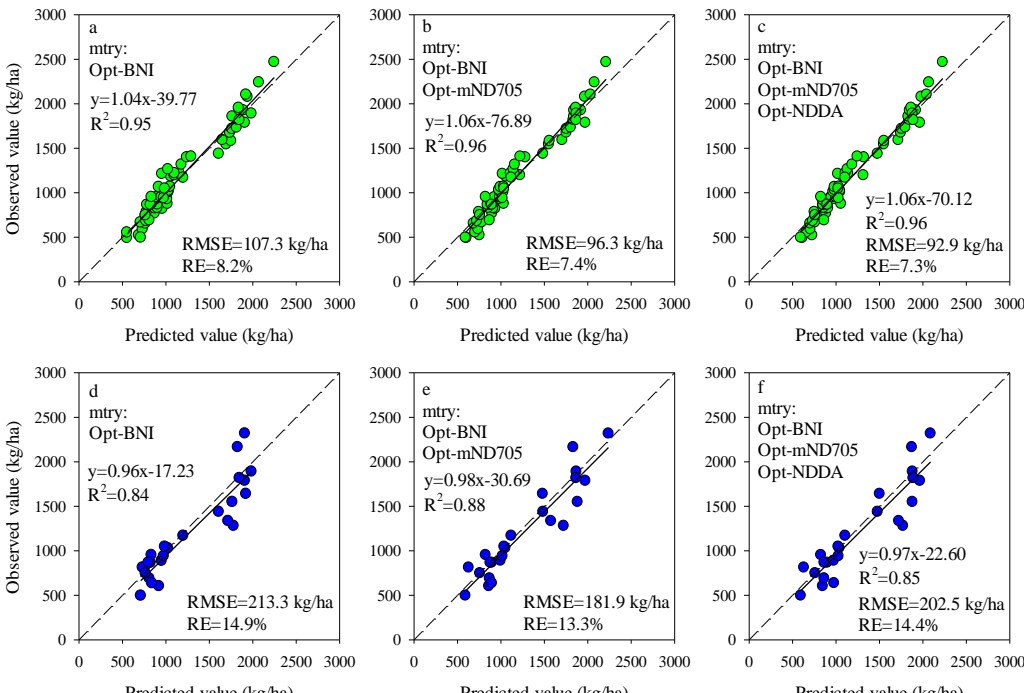

**Figure 9.** Optimizing the number of Opt-SIs predictors according to important scores at the tuber formation stage for training dataset (**a**–**c**) and testing dataset (**d**–**f**).

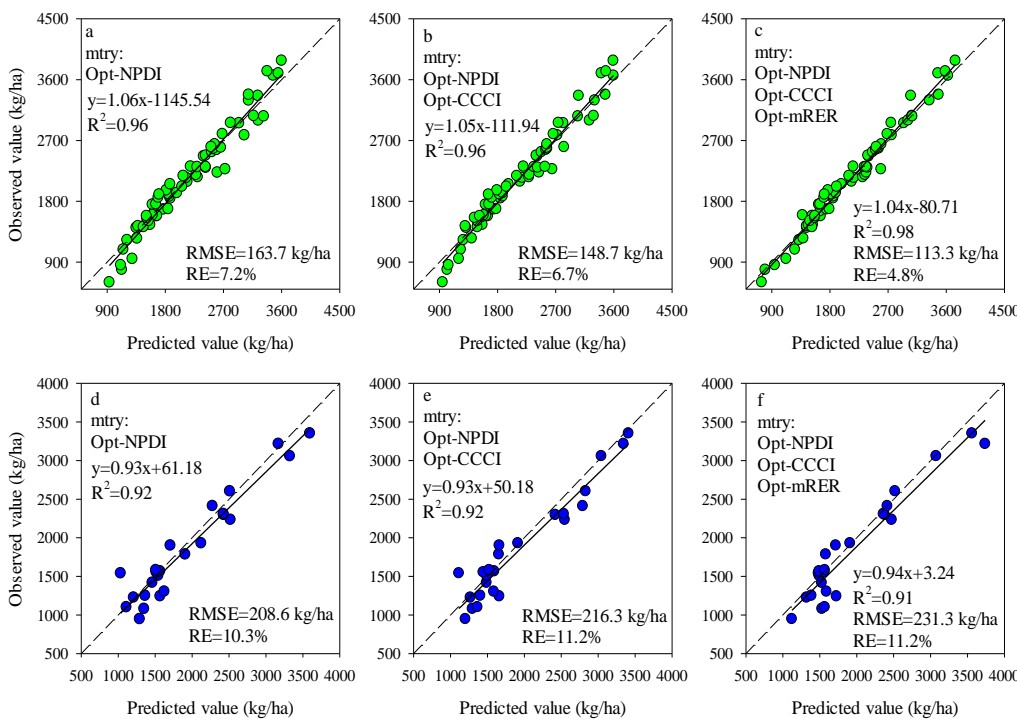

**Figure 10.** Optimizing the number of Opt-SIs predictors according to important scores at the tuber formation stage for training dataset (**a–c**) and testing dataset (**d–f**).

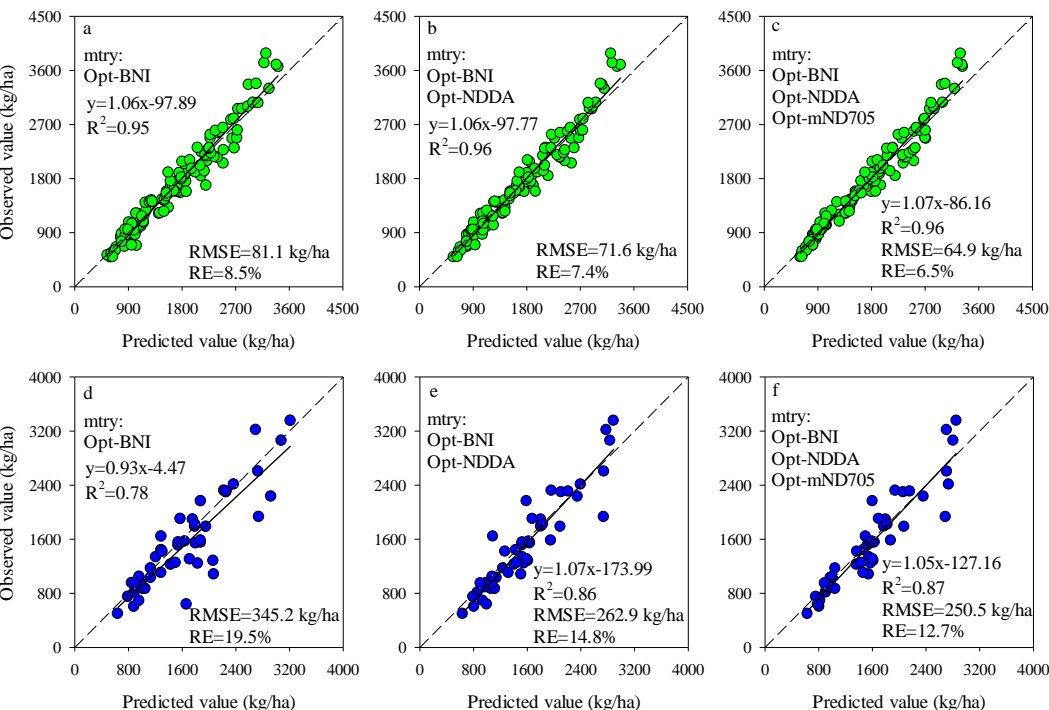

**Figure 11.** Optimizing the number of Opt-SIs predictors according to important scores at the combination of growth stage for training dataset (**a–c**) and testing dataset (**d–f**).

## 4. Discussion

### 4.1. The Performances of RF Models Coupling with Different Spectrum Variables

In this study, we investigated the RF model for the estimation of AGB in potato plants. Based on the canopy hyperspectral reflectance data, the full spectrum bands, published and Opt-SIs were compared for their performances of being used as predictors of RF models.

The types and numbers of input variables significantly influenced the performance of RF models in the estimation of potato AGB. The input variables using full-spectrum bands for the RF model had relatively high RMSE and RE for training and testing datasets in different stages. Compared with the full spectrum, the Pub-SIs coupling with the RF could improve by% 3–15% of the prediction ability in potato AGB at tuber bulking and the combination of two growth stages. Similar to the findings of Wang [41] and Niu [42], the machine learning models (RF, ANN, SVR, and MLR) coupling with spectral indices had better performances in predicting AGB at different growth stages. Nevertheless, the performances of different machine learning models can be different. For example, compared with ANN and SVR models, SIs combined with the RF algorithm had the best performances in predicting wheat AGB ($R^2 = 0.53−0.79$) [41]. Those results showed that SIs coupling with machine learning is a promising method to predict the AGB. In contrast, the results of Wang [6] suggested that the PLSR model coupling with Pub-SIs failed to improve the biomass estimation accuracy of maize AGB. One reason for the results may be that these Pub-SIs had high multicollinearity [6]. Another major cause might be that the original SIs used probably was insensitive to the estimation of maize AGB. In the current study, selecting 12 typical SIs and optimized the band combinations with different formula formats. Compared with the full spectrum bands and Pub-SIs, the Opt-SIs can significantly enhance the feature information. The Opt-SIs as input variables significantly increased the performance of RF algorithms in the estimation of potato AGB, suggesting that the Opt-SIs could potentially reduce the disturbance of growth stages and sites.

*4.2. The Comparison of Sensitive Bands*

Extracting sensitive bands from numerous spectral reflectance wavebands is very important for enhancing the prediction accuracy of SIs. At the tuber formation stages, the ultraviolet, violet and blue bands from 350–500 nm and NIR (900–1100 nm) are sensitive regions to potato AGB. However, the NIR (800–1100 nm) regions are important to AGB estimation in potato at tuber bulking and the combination of two growth stages. Similar to the findings of Fu [34], the NIR was the best sensitive region for winter wheat AGB. These areas are sensitive to dry matter and vegetation water content [65]. In contrast, many studies have indicated that the wavelengths in the red edge area contain useful information in the estimation of vegetation AGB [27,66–69]. For example, Kross [69] and Kanke [67] found that the red-edge SIs had better performance estimating crops AGB. Most of these results were reported in specific growth stages in wheat, corn, and rice, which suggest that the crop cultivars and growth stages have a great influence on the selection of sensitive bands. Therefore, the wavebands of SIs for the RF algorithms should be further optimized under different conditions to enhance the estimation accuracy.

Regarding the RF model based on full-spectrum bands, the sensitive bands were mainly located at the NIR radiation (1050–1150 nm) regions at the tuber formation stage. In the tuber bulking, however, the visible bands including ultraviolet (340–360 nm), blue (400–430 nm), and red edge (720–780 nm) were the most sensitive spectral regions to predict potato AGB. When two growth stages were combined, the spectral ranges from the red to red-edge (650–700 nm) and NIR (1080–1100 nm) were sensitive regions for the RF model to predict potato AGB (Figure 12). However, the findings of Dayananda [37] showed that the most important bands contributing to the RF method for biomass estimation were in the wavelength ranges of 546–910 nm (lablab), 750–794 nm (maize), and 686–814 nm (finger millet). These results confirm that the growth stages and crop cultivars have a great influence on the selection of sensitive bands.

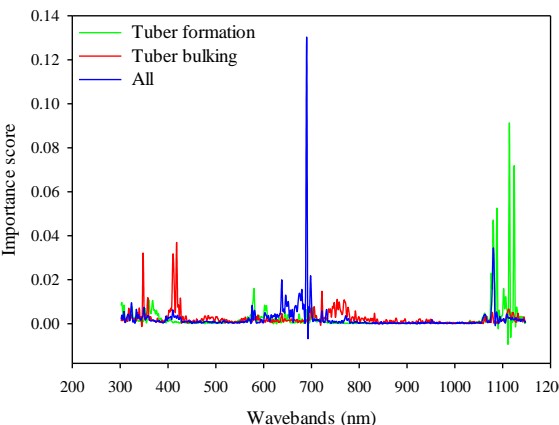

**Figure 12.** Comparison of the importance score for RF model based on full spectrum at different growth stages.

Compared with the full spectrum, the sensitive bands for the Opt-SIs as input variables were significantly different for each growth stage. One reason for the results is that the SIs combined several bands and specific formula formats to enhance spectral features sensitive to vegetation biochemical properties [70]. The wavebands of SIs contain some reference bands that are insensitive to AGB, and those bands are helpful to increase the signal-to-noise ratio [71]. Although the sensitive bands were different between the uses of Opt-SIs and full-spectrum, the sensitive spectral regions are consistent with some satellite bands, e.g., RapidEye and Sentinel-2A. It has been shown that the spectral indices and spectral bands from satellite image coupling with the artificial neural network algorithm could enhance the performance of the estimation model [72]. Therefore, those broad-band spectral reflectances or corresponding SIs coupling with machine learning algorithms are expected to improve the estimation of AGB on a large scale.

### 4.3. The Evaluation of Opt-SIs and RF Model

Existing studies have demonstrated that the growth stages have a significant influence on SIs performance in estimating the crops AGB [73,74]. In this study, the saturation effect appeared when the potato AGB was more than 2500 kg/ha (Figure 13). Similar to the current study, the saturation effect was found in wheat [33,34], maize [75], and rice AGB estimation based on SIs [15,76]. Therefore, many studies indicated that a single growth stage could improve the performance for using spectral indices to estimate the vegetable properties [49,77]. Compared to the best performing empirical models based on the optimized BNI, the Opt-SIs coupling with the RF algorithm can overcome the influences of saturation effect and significantly increase the estimation accuracy of potato AGB.

Following the variable selection and calibration, the most important step for RF is model optimization according to the predictor importance scores. It is particularly critical for reducing the probability of high-dimensional problems [78]. In the current study, the high prediction accuracy of potato AGB can be obtained by only using 1–3 Opt-SIs as the predictors of RF compared to the use of full-spectrum (424 bands), which was consistent across different growth stages. Similarly, Fu [34] found that the PLSR models based on optimized normalized difference vegetation index and optimized soil adjusted vegetation index and band parameters produced lower estimation errors (RMSE). Therefore, Opt-SIs provide an efficient way to reducing the high-dimensional problems of spectral analysis while enhancing the performance of machine learning.

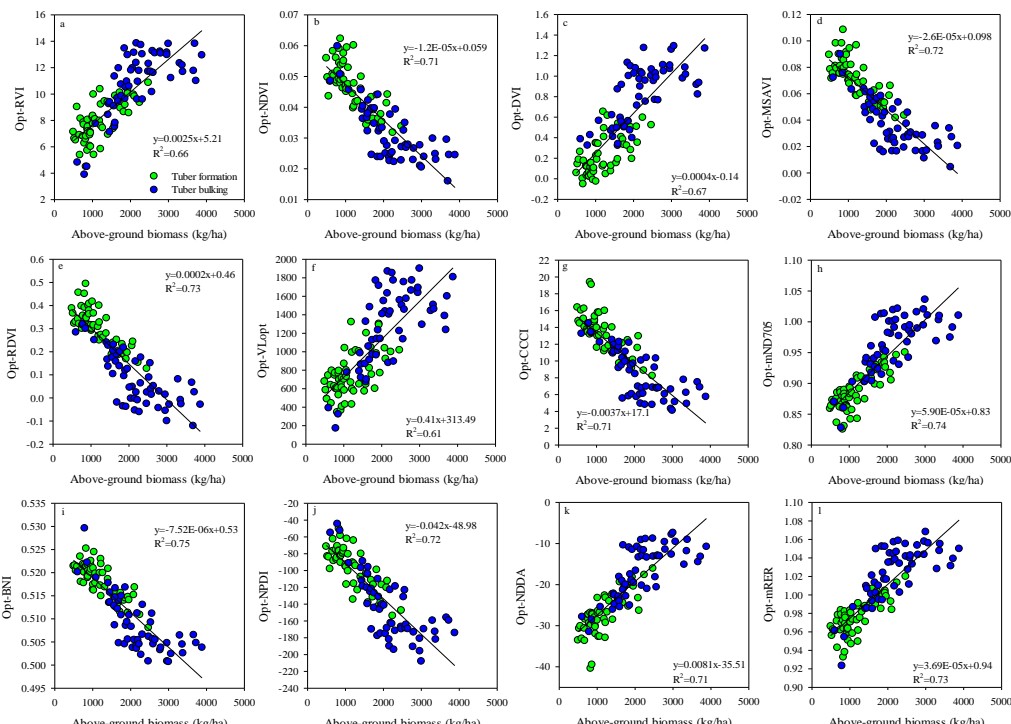

**Figure 13.** Relationships between potato above-ground biomass and (**a**) Opt-RVI, (**b**) Opt-NDVI, (**c**) Opt-DVI, (**d**) Opt-MSAVI, (**e**) Opt-RDVI, (**f**) Opt-VLopt, (**g**) Opt-CCCI, (**h**) Opt-mND705, (**i**) Opt-BNI, (**j**) Opt-NPDI (**k**) Opt-NDDA and (**l**) Opt-mRER at the combined growth stages.

## 5. Conclusions

To determine the most informative spectral predictors to be used in the RF model for the estimation of potato AGB, the full spectrum, Pub-SIs and Opt-SIs were used to train RF models using a dataset across growth stages. The Opt-SIs coupling with the RF model can significantly increase the potato AGB estimation accuracy for a single grow stage and across growth stages while significantly reduce the number of input variables. RF model based on the optimized BNI and NDDA could explain 86% of the variations in potato AGB. The Opt-SIs are promising predictors for training RF models for achieving robust and accurate estimation of crop biomass.

**Author Contributions:** Experiments were designed by F.L.; H.Y., and W.W. undertook the above-ground biomass extractions in the field; H.Y. compiled the data and performed the machine learning analysis; H.Y. wrote the initial draft of the manuscript and F.L. and K.Y. edited the manuscript. All authors have read and agreed to the published version of the manuscript.

**Funding:** This research was funded by Programs for Key Science and Technology Development of Inner Mongolia in 2019 and 2020 (2019GG248 and 2020GG0038), the National Natural Science Foundation of China (41361079), and Ph.D. research startup foundation of Inner Mongolia Agricultural University (BJ08-6).

**Data Availability Statement:** Not applicable.

**Conflicts of Interest:** The authors declare no conflict of interest.

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
