# Peer review of "Estimating Above-Ground Biomass of Potato Using Random Forest and Optimized Hyperspectral Indices"

_remotesensing, doi:10.3390/rs13122339_

Round 1
Reviewer 1 Report
The manuscript titled "Estimating above-ground biomass of potato using random forest algorithm from optimized hyperspectral indices" investigated to explore the random forest (RF) coupling with published spectral indices (Pub-SIs) and optimized spectral indices (Opt-SIs). The study showed that Opt-SIs wih RF algorithm enhance the prediction accuracy of crops above-ground biomass.
The ms fully falls within the editorial purposes of the journal and is focused on a very current issue, albeit extensively investigated in the literature. The ms is well structured and contains a large amount of very interesting and convincing data but an adequate review appears necessary before publication. I list some comments and suggestions for improvements below:
- Line 12-18: It is preferable to suffice with one sentence to define the problem in “Abstract”.
- The abstract did not have any values indicating the results.
- The keywords must not be repeated in the title.
- I suggest providing a figure to describe climatic conditions.
- Avoid to used “we” throughout the text.
- R2 need to add equation and statistical criteria need to add the missing references.
- Avoid to used “we” throughout the text.
- Graph 1:1 (Figs. 2, 4, 6, 8, 9 and 10) is preferred to compare the measured values with the modeled values, showing the regression equations and clarifying their coefficients (intercept and slop), besides the determination coefficient (R2).
- 5: Is random forest parameters ntree or mtry?
- The definition of abbreviations should be mentioned at the bottom of tables.
- The conclusions of the work appear for the most part as a mere repetition of the results and should be reduced by limiting themselves to the very significant considerations on you study.
Reviewer 2 Report
Remote sensing data is widely used to estimate various parametrs of agricultural crops to esimate their state, predict crop yield and production as well as to determine the fertilizer application rate. Machine learning techniques are efficient method to extract the relationships between remote sensing data and parameters of crop development. The article deals with the application of random forest algorithm to determine above-ground biomass of potato crop based on hyperspectral data. It analyses a number of spectral indices as well as tests random forest model optimization and how all it combined affects the accuracy of above-ground biomass prediction. The presented findings are valuable for the development of the efficient approach to operative and accurate estimation and monitoring of above-ground biomass of potato. However, there are some points requiring additional clarification or correction.
My comments can be found bellow.
The abstract needs correction. It should be self-explanatory. All the abbreviations should be explained when they mentioned first time: lines 23, 25.
Language check is also necessary:
line 21 - preposition is missing. what do you mean by explanantion ability of AGB?
line 36 - preposition is missing
line 37 - labor-meaurments ? Do you mean manual measurments?
line 60 - active voice is necessary: Many studies have demonstrated
Line 91 - some additional general explanantion on optimized spectral indices is necessary. it is not clear what exactly do you refer to in introduction.
line 138 - proper equation foramating is necessary
In Methods section, spectral indices should be described in separate subsection before the subsection on random forest regression. It would be more logical because first you calculate spectral indices than use it in random forest modelling.
Table 1 - footnote is necessary with the explanation of the parameters from columns Formulas and Algorithms
In tables 2 and 3 column "All" refers to the combination of two developmnet stages? If so, for consistency I suggest using the same word throughout the manuscript or providing a footnote to the tables with clarification. Especially as you are studying only two stages and not all of the development stages.
Line 431 - I suggest the correction of the statement or providing additional information in the manuscript as observed findings were not realated by the authors to N rates or years.
No information about soil of the test plots was presented in the manuscript. As the authors measured spectral reflectance of potato canopy, soil background can influence the informativeness of spectral indices used in the study especially those including visible spectral range. Applied N-rates can also be additional interferring factor as they influence the state of above-ground biomass and therefore its spectral reflectance.
The third aim of the research was to develop an efficient machine learning method for retrieving potato AGB. It was not mentined neither in results nor in the discussion section.
Reviewer 3 Report
Although the topic of the manuscript is of wide interest in remote sensing community, there are some issues the author may need to address.
1. Introduction
LL.79-85
There are a lot of machine learning algorithms and some of them showed the higher performances than RF. Your literature review is not enough.
2.2 data collection
L.142
Why did you choose this proportion?
Did you repeat the sampling procedure? It should be better for more robust conclusions.
Are all 79 samples collected in the experiment used to estimate AGB? Is there sample data available to verify the accuracy and stability of the model?
2.3 Random forest regression
LL.174-177
If the number of trees is increased, the generalization error always converges, and over-training is not a problem. Therefore, the reasons why you chose these settings were obscure.
Why did you ignore the other parameters?
Some studies showed that tuning the minimum number of unique cases in a terminal node, the maximum depth to which a tree should be grown and the number of random splitting were effective for improving accuracies.
You should have explained the importance score in the method section.
4. Discussion
LL.355-358
Why didn't you compare the estimation results among RF, SVM, ANN and PLS.
Round 2
Reviewer 3 Report
Although the authors improved the manuscript, there are some issues the authors may need to address in the revision.
1. Introduction LL.83-89
Could you offer more details on the previous studies? (e.g. accuracies, what SI did they use?)
2.4 Random forest regression
Was the prediction result determined by the majority vote?
Is this explanation for classification?
